# OpenReview forum: "Can LLMs Separate Instructions From Data? And What Do We Even Mean By That?"
_ICLR.cc/2025/Conference — ICLR 2025 Poster_

### Official Review · Reviewer_du3H · 2024-10-31

**Soundness:** 2
**Presentation:** 3
**Contribution:** 2
**Rating:** 6
**Confidence:** 4

**Summary:**

This paper explores the ability of large language models (LLMs) to distinguish between instructions and data within a given prompt. To evaluate this, the authors created a dataset where each sample contains {Task Prompt, Data Prompt, Probe Instruction, and Witness}. A perfect model would follow only the instructions from the task prompt while ignoring any instructions from the data prompt. If the model mistakenly follows the probe instruction, a "witness string" will appear in its output. By comparing the model’s behavior when the probe instruction is part of the task prompt versus when it appears in the data prompt, the authors assess its ability to separate instructions from data. Two evaluation metrics are introduced: the separation score and the utility score. The dataset and these metrics were used to evaluate GPT-3.5, GPT-4, and seven other models, ranging from 2B to 8B parameters. The paper also discusses three potential ways to improve model performance: prompt engineering, prompt optimization, and fine-tuning.

**Strengths:**

1. The paper is well-structured, with clear problem definitions, case studies, and experimental results.
2. The study is comprehensive, covering problem definition, evaluation metrics, dataset creation, experimental evaluation, and potential methods for performance improvement.
3. The dataset and reasonable metrics proposed provide an effective way to evaluate the instruction-data separation capabilities of LLMs.

**Weaknesses:**

1. One core contribution of the paper is the dataset; however, there are some questionable aspects regarding how it was built. As shown in Table 1, the "probe instruction" is appended to the end of the "data prompt," though they bear no semantic connection. Intuitively, this kind of example may not occur in real-world settings, creating input prompts that seem somewhat artificial. This raises concerns about whether the evaluation results truly reflect the model's ability to handle instruction-data separation in real-world usage. Moreover, the dataset creation process, as detailed in Appendix A, seems quite straightforward, being largely based on existing data and GPT-4, which furthers the aforementioned concern.
2. Some experimental setups and conclusions warrant more scrutiny:
    - **Experimental Setup**: In Tables 4 and 5, the baseline “Original” assigns the system prompt to the instruction argument, while the user prompt is treated as data. This setup seems problematic because it blends multiple instructions from the user and the system without clearly distinguishing what should be treated as instructions versus data, which may lead to input ambiguity. I suspect this confusion contributed to the low score for GPT-4 (20.8%) in Table 4. A more suitable baseline might be **PromptEng** method.
   - **Some conclusions appear misaligned with experimental results**:
       - In Line 450, the authors suggest fine-tuning significantly reduces utility, making it impractical. However, this conclusion seems premature. The poor performance of fine-tuning could be due to inadequate data quality or other subtle issues. Based on the current results, it’s too early to definitively state that fine-tuning is not a viable solution.
       - In Lines 461 and 497, the authors speculate that GPT-4's superior performance may be due to "principled differences in model architecture or training." However, the experimental data doesn’t robustly support this since models larger than 8B parameters were not included. Model size is a crucial factor that hasn’t been sufficiently considered. Including models like LLaMA3-70B, Qwen2.5-72B, or Mistral 8*7B would lend more solid support to the conclusions.

**Questions:**

1. What’s the primary difference between studying "separation of instructions and data" and "prompt injection"? Why is it important to study them separately? What are the potential consequences if we don’t?
2. This is an open-ended question to encourage the author to share their perspective. In what practical scenarios do you think the ability to separate instructions from data is especially critical? The paper doesn’t seem to delve deeply into this consideration.

---

> ### Author Response · Authors · 2024-11-23
>
> Thank you for your response. We’re glad you found our study well-structured and comprehensive. Below we address your concerns:
>
> **Fine-tuning and utility reduction in L450:** We did not mean to suggest that no way of fine-tuning could be a viable solution, only that in our experiments, it did not have the desired effect. In light of your comment, we substantially expanded fine-tuning in the revision. We now use a DPO objective that increases model utility by an average of 31.8%, making it more comparable to others. We've updated the text to reflect these new experiments.
>
> **Original vs PEng baseline:** We introduced the "Original" method to familiarize readers with our experimental design. We then acknowledge its shortcomings  and branch into three methods that reflect standard ways to address the problem. Any of these could be considered a baseline.  To better articulate this distinction, in the revision we rename Original to Naive and adjust the text accordingly. We'll also rethink Sections 5 and 6 structure and welcome your suggestions.
>
> **Probes appended to the end:** We use four different probe insertion methods (see Appendix D); Table 1 shows only one of them. We found out that position matters a lot, as e.g., putting the probe in the beginning of the user prompt increases the separation.
>
> **Artificial probes:** Intuitively, LLMs should achieve separation more easily when the probe is clearly unrelated to the main task. We made the setup as clear as possible to receive a strong signal about separation which is not muddled by e.g., injections.
>
> **Dataset creation:** Our goal in the dataset design was to make it easy to reproduce it and potentially scale it to larger sizes, while ensuring that an automatic and reproducible evaluation was possible (without, e.g., asking an LLM). As such, we see the relative simplicity of the dataset creation process as an advantage, not a shortcoming.
>
> **Clarifications on our statements regarding GPT-4’s superior performance:** We believe there was a misunderstanding due to our phrasing. In L461 we express our belief that today’s plain transformer architectures might not be well suited to ensuring instruction-data separation. In L497 we meant to express that we do not know why the GPTs behave rather well in our test, and we mention their architecture because that is not known to us, so it might be a reason. However, by using “architecture” twice we did not mean to suggest that GPT-4’s architecture is a desirable one to solve i/d-separation. Indeed, it could easily also be its scale, training data, or alignment process that influenced its results. We will change the wording to clarify this.
>
> **What’s the primary difference between studying "separation of instructions and data" and "prompt injection"? Why is it important to study them separately? What are the potential consequences if we don’t?**
>
> We believe these concepts operate on different levels: instruction-data separation is a property of the model, while prompt injection is a method of attacking models to compromise their security features. Prompt injections often succeed due to the fundamental lack of instruction-data separation in current models, causing them to execute input they should treat as data. Other factors like insufficient alignment might also contribute. Moreover, a lack of instruction-data separation is problematic even without an attacker. For example, in our email-processing scenario, harmless text out of context can lead to undesired behavior. To thoroughly understand prompt injections, we need to break them down into clear, well-defined issues like separation. Without this approach, we risk developing numerous brittle defenses that lead us nowhere.
>
> **… In what practical scenarios do you think the ability to separate instructions from data is especially critical?**
>
> Separating instructions from data is critical in security contexts, such as when models access sensitive data or can perform harmful actions. Similar to how CPUs and databases enforce instruction-data separation [1, 2], LLMs need this to be trustworthy.
> Retrieval-Augmented Generation (RAG) [3] applications benefit from separation. For instance, in Microsoft's competition involving LLMs processing potentially malicious emails [4], perfect separation would prevent executing harmful instructions. More generally, any setup involving data with natural instructions (e.g., summarizing dialogs) benefits from separation.
>
> [1] John L. Hennessy and David A. Patterson. Computer Architecture: A Quantitative Approach. Morgan Kaufmann, Publishers, 2017.
>
> [2] Justin Clarke-Salt. SQL injection attacks and defense. Elsevier, 2009.
>
> [3] De Stefano et al. Rag and Roll: An End-to-End Evaluation of Indirect Prompt Manipulations in LLM-based Application Frameworks. arXiv preprint arXiv:2408.05025, 2024.
>
> [4] Abdelnabi et al. LLMail-Inject: Adaptive Prompt Injection Challenge. Online article, 2024.

---

> > ### Comment · Reviewer_du3H · 2024-11-25
> >
> > Thank you for your response, which largely addresses my concerns. I'd like to change my rating from 5 to 6.

---

### Official Review · Reviewer_Z9dv · 2024-11-02

**Soundness:** 3
**Presentation:** 4
**Contribution:** 3
**Rating:** 8
**Confidence:** 3

**Summary:**

This paper motivates and formalizes the problem of instruction-data separation in LLMs - the ability to distinguish between instructions to be executed and data to be processed. The authors propose both a theoretical formal measure for instruction-data separation and a practical empirical metric for evaluating it. They introduce SEP, a carefully constructed dataset for testing instruction-data separation, and evaluate 9 popular LLMs using their methodology. Their results reveal that instruction-data separation is a significant problem in current LLMs, does not improve with model scale. These findings motivate the need for further research to address this limitation of LLMs.

**Strengths:**

- The results are properly caveated and presented with appropriate skepticism.
  - I appreciate that the authors explain their results with skepticism. E.g. pointing out that the results of GPT-4 may be impacted by the fact that GPT-4 created the SEP dataset (page 8); acknowledging that the set of prompt templates was not exhaustive (page 9); etc.

- Well written.
  - The paper was a pleasure to read. It was logical and easy to follow.
  - I appreciate that each definition or result has coherent discussion following it.
  - The problem of instruction-data separation is also well motivated.

**Weaknesses:**

- Some technical details are lacking.
  - See questions 1-3 below.

- Results are hard to make sense of.
  - As acknowledged by the authors, SEP performance varies widely between models (even between models of different scales from the same model family), as does the impact of the mitigations.
  - It is hard to draw conclusions from the results (Table 4, 5) as a result. The lack of clear patterns or trends makes it difficult to understand what factors contribute to better instruction-data separation in general.

**Questions:**

1. What was the fine-tuning training objective?
    - I am specifically wondering if there was a dual objective to both achieve good separability and also good utility, or if only one of these was incentivized in the fine-tuning procedure.

2. How were the "artificial" system prompts (used for Gemma and Starling) determined?
    - I'm wondering whether there was some trial and error / evaluation on some validation set to, in an effort to get a system prompt that behaved in a certain way. This (limited) optimization pressure could introduce some bias in the resulting "artificial" system prompt.

3. What is a task vs a subtask? (section 4)
    - In general I thought that the dataset creation methodology could have included more details.

---

> ### Author Response · Authors · 2024-11-23
>
> Thank you for your response. We're glad you enjoyed reading our paper and appreciated our discussions, motivation, and honesty about its limitations. Below, we address your questions.
>
> > What was the fine-tuning training objective?I am specifically wondering if there was a dual objective to both achieve good separability and also good utility, or if only one of these was incentivized in the fine-tuning procedure.
>
> Thank you for raising this point. Our initial fine-tuning used standard SFT with CrossEntropyLoss focused on separation. In response to your comment, we re-ran fine-tuning with three objectives: (1) SFT for separation, (2) DPO for separation, and (3) SFT balancing separation and utility. In the revision, we report DPO as our main result and provide additional analysis in Appendix B.4.
>
> >How were the "artificial" system prompts (used for Gemma and Starling) determined? I'm wondering whether there was some trial and error / evaluation on some validation set to, in an effort to get a system prompt that behaved in a certain way. This (limited) optimization pressure could introduce some bias in the resulting "artificial" system prompt.
>
> “Artificial” prompts were only used in the “Original” (now called “Naive”) experiment for Gemma and Starling, where we appended “System prompt” before the prompt. As noted, this method may not fully reveal i/d-separation, necessitating mitigation techniques. For prompt engineering, we performed validation by creating a dataset of 1,000 points differing from SEP (Appendix B.1), generating 16 pairs of system and data prompts (Appendix B.2), and calculating separation scores. We selected the best pair for each model to evaluate on the SEP dataset.
>
>
> >What is a task vs a subtask?
>
> A subtask is a specialized version of a task. For example, under “Part-of-Speech Tagging,” we used subtasks like “Adjective Identification,” “Noun Identification,” and “Conjunction Categorization.” A full list of 300 subtasks is in Appendix A.2.
>
> > In general I thought that the dataset creation methodology could have included more details.
>
> We already provide a detailed explanation in Appendix A, but if you prefer can also expand our description in the main body. Our publicly available code also includes thorough documentation and a step-by-step guide in the README for replicating our process or creating similar data.
>
> > Results are hard to make sense of. As acknowledged by the authors, SEP performance varies widely between models (even between models of different scales from the same model family), as does the impact of the mitigations.It is hard to draw conclusions from the results (Table 4, 5) as a result. The lack of clear patterns or trends makes it difficult to understand what factors contribute to better instruction-data separation in general.
>
> We explored patterns and trends in Appendix D. We found out that increasing instruction urgency counters LLMs' tendency to process instructions (Table 15); probe position significantly affects separation (Table 16); and that task domain matters: separation scores are highest for Information Processing tasks, followed by Analytical, then Creative tasks (Table 17). Regarding model size, we discuss in our revision that decreased separation in larger models may be due to increased task superposition [1]; smaller models struggle to execute both tasks.
>
> [1] Xiong et al. Everything Everywhere All at Once: LLMs can In-Context Learn Multiple Tasks in Superposition. arXiv preprint arxiv:2410.05603, 2024

---

> > ### Comment · Reviewer_Z9dv · 2024-11-25
> >
> > Thanks for the response, and for addressing my questions.
> >
> > Do you have a speculation as to why the fine-tuning reduces the utility measurement so much? What is the observed behavior of the resulting models? (I don't think the paper requires inclusion of this, but it would be nice to investigate and include in an appendix if it yields interesting observations.)

---

> ### Author Response · Authors · 2024-11-26
>
> Thank you for the question. The objective of fine-tuning is to prevent models from executing instructions within data. We observed that as an unintended consequence, the models become less likely to execute some of the instructions that should be executed (around a quarter of them, compared to the original). This is quite an interesting phenomenon, which might lead to valuable insights on fine-tuning. We'll investigate further and include any significant findings in an appendix in camera-ready.

---

### Official Review · Reviewer_m5E4 · 2024-11-04

**Soundness:** 2
**Presentation:** 3
**Contribution:** 2
**Rating:** 6
**Confidence:** 3

**Summary:**

This paper studies the problem of whether LLMs can separate instructions from data, which is important to the safety of LLMs. Specifically, this paper first introduces a formal measure for this problem, then proposes a new benchmark (i.e., SEP) to evaluate LLMs’ performance on this problem, and then conducts a study on the mitigation strategies of this problem.

**Strengths:**

- The paper explores an interesting and important research direction.
- The paper proposes a new benchmark, namely SEP, to evaluate the problem of instruction-data separation.

**Weaknesses:**

- There is a lack of detailed analysis on the evaluation results of different LLMs on SEP. For example, while authors report an abnormal phenomenon where better or larger models do not show stronger separation scores, they fail to provide either any detailed analysis or any explanation on the potential reason for this phenomenon.
- The study of mitigation strategies is not comprehensive. For example, while several existing fine-tuning techniques that target instruction-hijacking problems [1,2] can be naturally utilized to handle the problems in SEP, authors only include the vanilla fine-tuning technique in the study.

[1] Sizhe Chen, Julien Piet, Chawin Sitawarin, and David Wagner. StruQ: Defending against prompt injection with structured queries. arXiv preprint arXiv:2402.06363, 2024.

[2] Eric Wallace, Kai Xiao, Reimar Leike, Lilian Weng, Johannes Heidecke, and Alex Beutel. The instruction hierarchy: Training LLMs to prioritize privileged instructions. arXiv preprint arXiv:2404.13208, 2024.

**Questions:**

None beyond the weaknesses above.

---

> ### Author Response · Authors · 2024-11-23
>
> Thank you for your response. Below we provide clarifications to the questions you raised.
>
> > The study of mitigation strategies is not comprehensive. For example, authors only include the vanilla fine-tuning technique in the study.
>
> We have expanded our fine-tuning in the revised version. We performed multiple versions of fine-tuning with different objectives and reported the best results in our main table. Please see the main response for details.
>
>  > …while several existing fine-tuning techniques that target instruction-hijacking problems [1,2] can be naturally utilized to handle the problems in SEP…
>
> While existing fine-tuning techniques for prompt injections [1,2] could be applied to SEP, we are doubtful that including them would significantly impact our findings. For example, GPT-4o-mini was trained with instruction hierarchy [2] and was externally evaluated by [3] for prompt injections, achieving a 27% ASR compared to 48% for GPT-4o, still not resolving the issue. Nonetheless, in light of your suggestion, we performed additional fine-tuning using structured queries for Llama-3-8b, Llama-2-7b and Gemma-7b, increasing separation scores by an average of 1.96% for DPO and decreasing it by 0.53% for SFT. We discuss these results in Appendix E. Please note how getting a new number here doesn’t change the message of the paper.
>
> > There is a lack of detailed analysis on the evaluation results of different LLMs on SEP.
>
> We agree that detailed analysis of the behavior of specific LLMs is valuable but believe it is beyond this paper's scope. As the first work to study instruction-data separation in a principled way, our goals are to: (1) formally define the problem, (2) provide tools (benchmark and code), and (3) demonstrate its significance to the community.  We intend our work as a call for the LLM Safety community to focus on foundational problems like separation. We hope that in-depth analyses of specific models will become topics of future research.
>
> Note, that we already provide some ablation studies in Appendix D. We observe consistent patterns in separation scores based on data properties. Increasing instruction urgency counters LLMs' tendency to process instructions (Table 15). Probe position significantly affects separation (Table 16). Finally, task domain matters: separation scores are highest for Information Processing tasks, followed by Analytical tasks, and lowest for Creative tasks (Table 17).
>
> > For example, while authors report an abnormal phenomenon where better or larger models do not show stronger separation scores, they fail to provide either any detailed analysis or any explanation on the potential reason for this phenomenon.
>
> Thank you for highlighting this. We have added a discussion in the revised manuscript. We believe that smaller models show higher separation because they cannot execute both tasks simultaneously, whereas larger LMs are better at task superposition [1] and tend to execute both. This suggests i/d-separation is a pressing issue for larger LLMs.
>
> [1] Xiong et al. Everything Everywhere All at Once: LLMs can In-Context Learn Multiple Tasks in Superposition. arXiv preprint arxiv:2410.05603, 2024
>
> [2] Wallace et al. Training LLMs to prioritize privileged instructions. arXiv preprint arXiv:2404.13208, 2024
>
> [3] Debenedetti et al. AgentDojo: A Dynamic Environment to Evaluate Attacks and Defenses for LLM Agents. NeurIPS D&B, 2024

---

> > ### Comment · Reviewer_m5E4 · 2024-12-02
> >
> > Thank you for the additional experiments and discussion. I have raised my score from 5 to 6.

---

### Author Response · Authors · 2024-11-23

We thank the reviewers for their feedback. We are encouraged that they found the topic of our work (m5E4: *“interesting and important”*, Z9dv: “well motivated”), appreciated the exposition (Z9dv: *“pleasure to read”*; du3H: *“well-structured”*) and found positive words about the dataset and evaluation metrics that are our main contribution (du3H: *“effective way to evaluate the instruction-data separation capabilities of LLMs”*). We also thank the reviewers for their constructive suggestions, which we have incorporated into a revised paper with improved experiments and discussions. We address individual questions in separate responses and use this comment to provide an overview of the changes in the revision.

**Expanded fine-tuning experiments**

In our experiments, fine-tuning achieved the highest separation score but significantly reduced model utility, making it impractical. Reviewers m5E4, du3H, and Z9dv asked about its simplicity, potential training issues, and whether we used a dual objective to balance separation and utility. We appreciate these suggestions and have addressed them in our revision:

1) We now fine-tuned models with three objectives: (1) original Supervised Fine-Tuning (SFT), (2) Direct Preference Optimization (DPO) on pairs of probe and no-probe data, and (3) SFT with a dual objective for both separation and utility.

2) We expanded hyperparameter search for the phi-3 and gemma models, which had lower scores initially in the original experiment.

We observed substantially increased separation for phi-3 and gemma-7b with SFT, making them comparable to other models, which also received a minor improvement (1-2%). Due to improvements in phi-3 and gemma, SFT's average separation increased from 81.8% to 94.4% and utility increased from 19.2% to 47%, though still well below other methods (62.2 -- 69.9%). On average, DPO slightly outperformed SFT in both separation (1.5% higher) and utility (2.3% higher). Double-objective training achieved 94.4% average separation and 47.7% average utility. Overall, we find the results quite consistent.
We now report DPO results in the main text due to its higher utility and practicality. Full fine-tuning experimental results are provided in Appendix B.4.
We have also expanded the discussion on fine-tuning in the manuscript accordingly.

**Improved discussion**

In the revision, we clarified statements that reviewers had asked for and added more discussions where needed. Specifically, we now discuss why larger models might have lower separation (m5E4, Z9dv), clarify our statement on GPT-4's performance (du3H), and rename our "Original" method to "Naive" to articulate its role in the discussion as an introduction to other mitigation techniques.

---

### Meta-Review · Area_Chair_zVSg · 2024-12-20

**Metareview:**

The paper studies the instruction-data separation problem in LLMs. It introduces a formal measure for this problem, proposes a new benchmark to evaluate the performance of LLMs on this problem, and suggests mitigation strategies for this problem.

+ The paper is well-written.
+ The experiments are comprehensive.

- Some of the results require more analysis.

**Additional Comments On Reviewer Discussion:**

Some issues raised in the initial reviews were sufficiently addressed in the rebuttal. This includes additional experiments, an analysis of the results, as well as discussion on the technical details that were considered to lack sufficient clarity by the reviewers. Given the clarifications provided in the rebuttal, all reviewers recommend acceptance.

---

### Decision · Program_Chairs · 2025-01-22

Accept (Poster)